# Mechanistic Insights into Selective Autophagy Subtypes in Alzheimer’s Disease

**DOI:** 10.3390/ijms23073609

**Published:** 2022-03-25

**Authors:** Xinjie Guan, Ashok Iyaswamy, Sravan Gopalkrishnashetty Sreenivasmurthy, Chengfu Su, Zhou Zhu, Jia Liu, Yuxuan Kan, King-Ho Cheung, Jiahong Lu, Jieqiong Tan, Min Li

**Affiliations:** 1Mr. & Mrs. Ko Chi-Ming Centre for Parkinson’s Disease Research, School of Chinese Medicine, Hong Kong Baptist University, Hong Kong, China; 21481555@life.hkbu.edu.hk (X.G.); iashok@hkbu.edu.hk (A.I.); sravangs@hkbu.edu.hk (S.G.S.); 20481403@life.hkbu.edu.hk (C.S.); zhuzhou1@hkbu.edu.hk (Z.Z.); 20482248@life.hkbu.edu.hk (J.L.); yuxuan@hkbu.edu.hk (Y.K.); kingho@hkbu.edu.hk (K.-H.C.); 2Institute for Research and Continuing Education, Hong Kong Baptist University, Shenzhen 518057, China; 3State Key Lab of Quality Research in Chinese Medicine, University of Macau, Macao, China; jiahonglu@um.edu.mo; 4Center for Medical Genetics, School of Life Sciences, Central South University, Changsha 410000, China

**Keywords:** Alzheimer’s disease, selective autophagy, aggrephagy, mitophagy, reticulophagy, lipophagy, pexophagy, nucleophagy, ribophagy, lysophagy

## Abstract

Eukaryotic cells possess a plethora of regulatory mechanisms to maintain homeostasis and ensure proper biochemical functionality. Autophagy, a central, conserved self-consuming process of the cell, ensures the timely degradation of damaged cellular components. Several studies have demonstrated the important roles of autophagy activation in mitigating neurodegenerative diseases, especially Alzheimer’s disease (AD). However, surprisingly, activation of macroautophagy has not shown clinical efficacy. Hence, alternative strategies are urgently needed for AD therapy. In recent years, selective autophagy has been reported to be involved in AD pathology, and different subtypes have been identified, such as aggrephagy, mitophagy, reticulophagy, lipophagy, pexophagy, nucleophagy, lysophagy and ribophagy. By clarifying the underlying mechanisms governing these various subtypes, we may come to understand how to control autophagy to treat AD. In this review, we summarize the latest findings concerning the role of selective autophagy in the pathogenesis of AD. The evidence overwhelmingly suggests that selective autophagy is an active mechanism in AD pathology, and that regulating selective autophagy would be an effective strategy for controlling this pathogenesis.

## 1. Introduction

Alzheimer’s disease (AD) is the most prevalent progressive neurologic disorder affecting aged people worldwide. Clinically, the primary manifestation of AD is memory impairment, followed by deficits in cognitive and adaptive functioning as the disease progresses [1,2,3]. On the basis of several factors, AD can be broadly categorized as familial and sporadic AD. Familial AD, also known as early onset AD, is often associated with genetic mutations, such as mutations in amyloid precursor protein (APP), presenilin 1 (PSEN1), and presenilin 2 (PSEN2). Sporadic AD is often caused by environmental and complex genetic factors such as the presence of the epsilon4 allele in apolipoprotein E (APOE) [4]. Pathologically, AD is characterized by the accumulation of tau-enriched neurofibrillary tangles (NFTs) and β-amyloid (Aβ) toxic aggregates in the hippocampus and cerebral cortex of the brain [5], which are the major pathological markers of AD [6,7]. According to previous studies, an accumulation of NFTs leads to synaptic dysfunction, which further causes a decline in memory and cognitive functions. Moreover, Aβ oligomers disrupt calcium homeostasis by binding to receptors, activating calcineurin and caspases [8]. In AD patients, a large number of aggregated toxic proteins interfere with the normal function of autophagy, further aggravating the pathogenesis of AD. Increasing evidence has suggested that autophagy, through which damaged organelles and misfolded proteins are degraded and then recycled to maintain proteostasis, plays a major role in AD. The mammalian nervous system, especially the neurons, is largely dependent on autophagy to clear large amounts of insoluble protein aggregates to maintain protein homeostasis [9]. Initially, autophagosomes and autolysosomes were found to accumulate in AD brains [10]. Early studies demonstrated that Beclin-1, a core subunit of the vacuolar protein-sorting 34 (VPS34) complex, decreases as AD progresses [11,12,13]. Interestingly, phosphatidylinositol 3-phosphate (PI3P) production, mediated by the VPS34 complex, is also downregulated in AD patients’ brains [14]. Autophagy is a protective mechanism in response to stress in the early stages of AD, but autophagic flux is reduced in the late stages of AD [15]. In our previous studies, we found that many Chinese herbs, formulas and compounds have neuroprotective effects and can promote the clearance of Aβ and tau proteins through enhancing autophagy and rescuing memory and cognitive impairment in AD mouse models [16,17,18,19,20]. Mounting evidence is leading to the conclusion that dysfunctional autophagy is strongly linked to AD pathogenesis.

In neurodegenerative diseases, especially AD, activating autophagy can promote cell survival by removing damaged organelles and protein deposits. Autophagy thereby provides additional energy, and dictates the fate and function of various cellular organelles [21]. According to the different pathways by which the cargo can be degraded in lysosomes, autophagy can be divided into three types: macroautophagy, microautophagy, and molecular chaperone-mediated autophagy (CMA). In macroautophagy, the double-membrane autophagosome wraps the cargo to be degraded, fuses with the lysosome, and degrades in the lysosome; this is the most common form of autophagy [22]. In conditions of nutrient imbalance, autophagy is generally a nonselective cellular degradation process. However, damaged and excess cellular organelles such as mitochondria, ER, lysosomes, and protein aggregates can also be degraded in a highly selective manner through selective autophagy [23]. For this type of autophagy, some autophagy-related proteins are required to ensure specific recognition and degradation of the cargo [21]. For example, the autophagy-related protein 8 (Atg8) family proteins on autophagosomes can target specific proteins for degradation. Microautophagy refers to the transport of cargo into lysosomes through lysosomal membrane invagination for degradation [24]. CMA refers to the process in which the cargo to be degraded binds to molecular chaperones and directly targets the lysosomes for degradation [25]. Heat shock cognate protein 70 (Hsc70) transports the target protein to the lysosomal receptor for degradation [26]. Ubiquitination is a marker of targeted protein degradation [27]. A variety of selective autophagy processes have been discovered and named according to their cellular targets, such as mitophagy (mitochondria), reticulophagy (the ER), lipophagy (lipid droplets), pexophagy (the peroxisome), nucleophagy (the nucleus), ribophagy (the ribosome), and lysophagy (the lysosome). All of the abovementioned selective autophagy subtypes are essential for maintaining proper cellular homeostasis [28].

Selective autophagy dysfunction is correlated with AD pathogenesis [29,30]. Aggrephagy and mitophagy dysfunction have been shown in the clearance of aggregated proteins and damage mitochondrial in AD patients [31]. There is some research on the link between AD and reticulophagy, which indicates that reticulon 3 (RTN3) aggregation, interferes with reticulophagy, and increases Aβ levels in elderly people [32]. Moreover, the presence of large amounts of lipid droplets (LDs), such as cholesteryl esters in the neurons, reduces the activity of the proteasomes, thereby promoting the accumulation of p-tau protein, further contributing to the pathogenesis of AD [33]. Therefore, clarifying the mechanism of selective autophagy in the pathogenesis of AD may lead us to a strategy for the prevention and treatment of AD. In this review, we summarize the current knowledge on the mechanisms of the various subtypes of selective autophagy in AD pathogenesis.

## 2. Aggrephagy

Generally, toxic aggregates cause neuronal membrane permeability damage, calcium homeostasis dysfunction, inflammation, neurotoxicity, and physiological abnormalities [34,35,36]. Aggrephagy is a subtype of selective autophagy that clears toxic protein aggregates [37]; in that sense, it reduces AD pathology. Aggregates are moved into double-membrane autophagosomes, where they are degraded in the acidic lysosomal lumen. The receptor SQSTM1-p62 (p62), the neighbor of BRCA1 gene 1 (NBR1), and the large junction protein ALFY are important in aggrephagy. A deficiency of ALFY or p62 is associated with neurodegeneration [38,39]. Autophagic receptors such as NBR1 and p62 directly bind to Atg8 homologs on autophagosomes for the degradation of protein aggregates [40,41,42,43,44]. Furthermore, CMA and the ubiquitin–proteasome system (UPS) prevent increases in soluble and insoluble protein aggregation [45]. Heat shock proteins such as Hsp40, Hsp60, Hsp70, and Hsp90 bind to the misfolded protein, neutralize hydrophobic areas exposed by the misfolded proteins, and prevent aggregation [46]. In CMA, Hsc70 binds to the KFERQ motif of the misfolded protein and is delivered to lysosomes for degradation by LAMP2a [47]. Ubiquitination of aggregates is critical for degradation in UPS and classical autophagy systems [48]. Existing studies have shown that ubiquitin interacts with seven lysine residues (K6, K11, K27, K29, K33, K48, and K63) [49]. These seven lysine residues bind with aggregates covalently, forming polyubiquitin chains for degradation [50].

AD is a proteinopathy, with the aggregation of Aβ and P-tau protein in the brain leading to memory deficits and neurodegeneration. The elevated Aβ-42 monomers gradually assemble into aggregates within the perinuclear space [51]. Normally, ubiquitin protein binds with Aβ for UPS degradation [52]. However, these polyubiquitin chains are unstable when bound with Aβ. Therefore, they cannot fold properly, and Aβ folding dysfunction changes their physicochemical properties and promotes the formation of Aβ fibers and plaques. Overall, due to polyubiquitination, Aβ is unfolded, causing the unfolded Aβ aggregates to develop into insoluble Aβ fibers. During aggregation, some polyubiquitin Aβ peptides are degraded normally by autophagy. Other polyubiquitin Aβ fibers and plaques bind to the UBA domains of p62 and NBR1 selectively, initiating degradation by aggrephagy [53].

Clinically, in the brains of AD patients, an enormous amount of extracellular Aβ and intracellular neurofibrillary tangles can be found [54]. NFTs are formed by abnormal aggregation and hyperphosphorylation of tau into paired helical filaments (PHFs) [55]. P-tau is transported to the nucleus to form aggregates for lysosomal degradation [56]. When proteasomal aggregation degradation is inhibited, Hsc70 interacts with CHIP to regulate tau ubiquitination [57]. Ubiquitinated P-tau is recruited by the aggregation receptor p62 [56,58]. HDAC6 binds with p62 and enhances retrograde transport of the P-tau and p62 complex to form tau aggregates [59]. Furthermore, HDAC6 has been reported to respond to the protease imbalance induced by proteasome inhibition and recombine with the cytoskeletal actin network to induce the formation of tau aggregates [60]. It has been reported that the ubiquitin carboxyl-terminal hydrolase L1 (UCHL1) enzyme also affects the formation of tau protein aggregates. UCHL1 is a deubiquitinating enzyme that is abundantly expressed in the brain [61]. UCHL1 inhibition leads to hyperphosphorylation of tau protein and tau microtubule-stabilizing dysfunction [62]. In addition, UCHL1 can regulate the activity of HDAC6 and affect the formation of tau protein aggregates. Inhibition of UCHL1 impairs the interaction between P-tau and HDAC6, inhibiting the formation of tau aggregates [63]. Undoubtedly, specific aggregate degradation is required to prevent AD progression. Figure 1 illustrates the toxic protein aggregates degraded by the mechanism of aggrephagy in AD.

## 3. Mitophagy

Mitochondria are critical for synaptic transmission, synaptic plasticity, calcium homeostasis, and neuronal survival [64,65]. Damaged mitochondria are degraded through mitophagy [66,67]. Impaired mitophagy is associated with a variety of neurodegenerative diseases, including AD [28]. Mitochondrial dysfunction appears with age, but it is exacerbated in AD [68,69,70,71]. Mitochondria are key targets of autophagy in AD patient brains [72,73]. In AD models, mitochondrial swelling, abnormal morphology, and the accumulation of large numbers of damaged mitochondria have been found [74,75,76]. For example, when the mitochondria are damaged, the membrane potential decreases, which stabilizes PINK1 to the outer membrane. PINK1 phosphorylates and recruits the E3 ubiquitin ligase Parkin to the mitochondria, and mediates the ubiquitination of mitochondrial outer membrane proteins to mark impaired mitochondria for degradation through mitophagy [77]. Increased PINK1, Parkin, and mitochondrial protein ubiquitination were detected in an AD mouse model, suggesting that the mitophagy process is activated but stalled [78]. This may be due to an insufficient capacity to remove many damaged mitochondria, thereby resulting in the accumulation of impaired mitochondria and disturbance of mitochondrial homeostasis. In addition, corresponding to the age-related increase in AD incidence, mitochondrial dysfunction and mitophagy also increase with age [79].

Impairment of the mitochondria occurs in the early stages of AD, and this impairment can even appear before Aβ and NTFs are formed [80,81,82,83]. Decreases in oxidative phosphorylation, an increase in ROS production, and downregulation of the tricarboxylic acid cycle (TCA) have been reported [84,85]. Studies have found that the energy metabolism in AD patients’ brains is lower than normal, which may be due to mitochondrial dysfunction [86]. PGC-1-alpha, a major regulator of mitochondrial biogenesis, is decreased in AD patients’ brains, indicating mitochondrial dysfunction [87,88]. In addition, the accumulation of damaged mitochondria has been confirmed in the neurons of AD brains. This may be due to severe impairment of the clearance of damaged mitochondria (such as through mitophagy) [89,90]. Mitophagy dysfunction can cause the accumulation of autophagic vacuoles (AVs) in neuronal cell bodies. AVs contain a large number of Aβ polypeptides, and they are a location where Aβ easily forms aggregates [91]. In AD patients, we found that Parkin and PINK1 are reduced, which decreases the occurrence of mitophagy, and the damaged mitochondria cannot be cleared [92], causing a dysfunction of mitochondrial transport, hyperphosphorylation of tau protein, and synaptic dysfunction [93].

Aβ/tau is also involved in mitochondrial dysfunction in AD. Overexpression of APP in the neurons leads to mitochondrial dysfunction [89]. Aβ and P-tau can directly interact with mitochondrial components, such as VDAC1 and Complex IV, and interfere with their function [94,95,96]. Mitochondrial dysfunction has been found in AD patients and mouse models; specifically, mitochondrial membrane potential is decreased, ROS production is increased, and mitochondrial swelling occurs [97,98,99]. In an AD cell model, mitofusin 1/2 (MFN1/2) decreased and Aβ oligomers induced mitochondrial fragmentation. Further studies have shown that Aβ oligomer mediates the activation of dynamin-related protein 1 (Drp1); this mitochondrial division proteins cause mitochondrial fragmentation and cell death in AD patients [100]. In addition, extracytoplasmic calcium is harmful to neurons and its presence may lead to neuronal death [101]. The endoplasmic reticulum and mitochondria are the main locations where intracellular calcium is stored, and they play a key role in intracellular calcium homeostasis [102,103]. It has been reported that Aβ oligomers disrupt intracellular calcium homeostasis. Aβ enhances glutamate neurotransmission and increases tau phosphorylation through n-methyl-D-aspartate receptors (NM-DARs), resulting in impaired mitochondrial function [104,105,106]. Tau pathology is also involved in AD mitochondrial dysfunction [107,108,109,110,111,112,113,114]. Studies have reported that P-tau leads to downregulation of Optic Atrophy 1 (Opa1) and Drp1. In addition, P-tau may cause an increase in mitochondrial membrane potential, thereby preventing the mitochondrial recruitment of Parkin by PINK1 [107,115,116,117]. In AD, hyperphosphorylation of tau can influence tau’s ability to bind to microtubules and can disrupt mitochondrial axon transport [118,119]. Mitochondrial fusion gene mutations have been shown to lead to mitochondrial fragmentation and AD pathology [120]. In addition, an inhibitor of mitochondrial Complex I induced tau protein pathology in rats [121,122]. Studies have shown that oxidative stress increases γ-secretase activity [123]. Similarly, excessive ROS promotes P-tau aggregation through phospholipid peroxidation [124,125]. Therefore, the accumulation of Aβ and p-tau proteins can aggravate mitochondrial dysfunction, and mitochondrial dysfunction can also aggravate tau phosphorylation and Aβ aggregation. In fact, there may be a vicious cycle between mitochondrial dysfunction and AD pathology. Figure 2 illustrates the key events in mitophagy as it relates to AD.

## 4. Reticulophagy

The endoplasmic reticulum (ER) is a network that is responsible for lipid and protein biosynthesis, and the modification and folding of secreted proteins. There are two vital quality control mechanisms in the cell, namely the unfolded protein response (UPR) and the ER-associated degradation pathways (ERAD) [126,127]. Activation of UPR has been reported to induce the formation of ER-associated autophagosomes, which can specifically wrap the ER and then fuse with the lysosome for degradation [128,129]. Reticulophagy seems to coexist with the downstream signaling pathways of ER stress, balancing the expansion of the ER [130,131,132]. Studies have shown that a PS1 mutation inhibits IRE1 signaling. IRE1 is an ER stress sensor serine/threonine kinase involved in AD pathogenesis, in which the IRE1α/JKN pathway may predispose a subject towards amyloid deposition and the process of AD [133,134]. Transcription factor XBP1 can bind to the promoter region of γ-secretase complex (*Psen1 and Ncstn*), which play an important role in Aβ synthesis [135]. Studies have shown that phospho-PERK increased in AD patients and that PS1 mutants inhibited PERK signaling [133,136]. Several receptors of reticulophagy have been discovered: FAM134, SEC62, RTN3, and CCPG1. RTN3 is involved in the development of neurodegenerative diseases [32]. RTN3 has several cleavage isoforms, the longest of which has six LIR domains that can bind with LC3/GABARAP and enhance the segmentation of ER tubules, inducing reticulophagy. In fact, RTN3 mainly initiates reticulophagy under conditions of deficient energy or hypoxia. RTN3 is a protein containing RHD that is located in the ER tubules, and its main function is to promote the formation of more ER tubules [137]. RTN3 was initially identified as a negative regulator of BACE1. BACE1 is an enzyme that cleaves APP to release beta-amyloid peptides. Increased RTN3 monomer expression affects the intracellular transport of BACE1, resulting in more BACE1 being retained in the ER, where the cleavage capacity of BACE1 for APP is attenuated. Interestingly, RTN3 can form aggregates, and an increase in the number of aggregates of RTN3 appears to be related to the formation of RTN3 immunoreactive dystrophic neurites in AD patients’ brains. Therefore, RTN3 aggregation counteracts the normal function of RTN3, by negatively regulating the RTN3 monomer-mediated BACE1 activity [32].

Growing evidence has suggested that UPR activation markers are increased in AD patients’ brains and animal models [136,138]. Studies have shown that the deposition of Aβ attenuates the interaction of the ER and microtubules in the hippocampal neurons, leading to the activation of reticulophagy [139]. In addition, chemicals that interfere with the cholesterol metabolism in the ER have been reported to increase the clearance of Aβ42, further suggesting that ER dysfunction is closely related to AD [140]. In addition, ER stress triggers autophagy, greatly decreasing APP and amyloid beta precursor-like protein 1 (APLP1). The inhibition of autophagy contributes to the deposition of APP and APLP1 [29]. Furthermore, tau has contributed to our understanding of the relationship between ER and autophagy. A study found that ER stress induces autophagy to reduce the deleterious effects of tau [141]. Therefore, the relationship between reticulophagy and AD needs further exploration. Figure 3 depicts the mechanism by which reticulophagy appears to be involved in aging and AD.

## 5. Lipophagys

Lipid droplets (LDs) are organelles that store lipid. Mitochondria and the ER respond to ER stress and provide mitochondrial energy [142]. Lipophagy is a subtype of selective autophagy that targets LDs for degradation and plays a role in regulating lipid storage. In lipophagy, LC3-positive phagocytes engulf LDs and gradually form autophagosomes, fusing with lysosomes and degrading to glycerol and free fatty acids. Different LD-resident proteins participate in targeting LDs to undergo lipophagy, including the molecular chaperones, the Rab molecular switch family, and the UPS system [143]. In addition, adipose triglyceride lipase (ATGL) has been demonstrated to regulate lipophagy by interacting with LC3 [144]. LDs can be directly targeted for degradation by lysosomes. Furthermore, the LD-associated protein perilipins can be degraded by CMA, making LDs more susceptible to lipophagy and lipolytic degradation [145].

Many studies have shown that there is a relationship between AD and autophagy. The accumulation of LDs has been detected in mouse models of AD and the human brain [146]. There is ample evidence that LD accumulation is correlated with the pathology and severity of AD. When neurons are overactive or undergo oxidative stress, LDs accumulate in the neurons and are then transported to the astrocytes for degradation [147]. Interestingly, LD accumulation in the astrocytes has been shown to reduce the impact of ROS, thereby providing an antioxidant defense mechanism [148]. Similarly, the increased ROS production caused by neuronal dysfunction promotes the accumulation of LD in the microglia [149]. Although LDs in the microglia were initially thought to have a protective effect, it has been shown that an accumulation of LDs in the microglia can induce oxidative stress in mouse models [150]. Providing further evidence for the destructive role of LD accumulation in AD, studies have shown that cholesteryl esters in LDs reduce proteasome activity, therefore promoting the accumulation of P-tau [33]. Furthermore, studies have shown that the accumulation of LDs in AD mouse models inhibits the regenerative and homeostatic functions of neural stem cells [146]. There are a large number of LDs in neurons, and the normal metabolism of these lipids is essential for maintaining healthy neuronal function. Several studies have shown that a reduction in the cholesterol content of neurons greatly affects neuronal activity and neurotransmission, resulting in progressive degeneration of dendritic spines and synapses, which is the main characteristic of AD pathogenesis [151,152,153]. Furthermore, sphingomyelinase has been shown to increase neuronal apoptosis by producing the pro-apoptotic molecule ceramide [154,155,156]. Therefore, given the critical role of LD accumulation in AD pathology, lipophagy may represent a novel therapeutic strategy. Indeed, choline supplementation has been explored as a novel AD therapy [157]. Given this evidence, further investigation of the relationship between lipophagy and AD is warranted.

## 6. Pexophagy

Peroxisome dysfunction appears to be associated with many neurodegenerative diseases, including AD and Parkinson’s disease (PD) [158]. Peroxisomes include enzymes such as catalase (which can break down ROS), glutathione peroxidase (which breaks down hydrogen peroxide), and superoxide dismutase (which breaks down superoxide) [159]. Furthermore, they have been implicated in adipogenesis and ROS signaling in the heart and gut [160,161]. In particular, peroxisomes synthesize the lipids that make up myelin sheaths, cell membranes, and other phospholipids in the neurons. Peroxisome dysfunction impairs neuronal migration and cell membrane synthesis [162,163,164,165].

Pexophagy refers to selective degradation of the peroxisomes by autophagy. Thus, it controls the quality of the peroxisomes [166]. However, pexophagy-specific receptors have not been well studied in mammalian cells. Pexophagy mainly relies on the ubiquitination of peroxisomal proteins in mammals [167,168]. Studies have shown that overexpression of NBR1 and p62 induces peroxisome clustering and degradation. P62 is indispensable for pexophagy, whereas NBR1 overexpression manifests as hyperactivation of pexophagy. p62-NBR1 binding increases the efficiency of NBR1-mediated pexophagy [167]. Studies have shown that peroxisomal biogenesis factor 5 (PEX5) must be ubiquitinated by peroxisomal biogenesis factor (PEX2), an E3 ubiquitin ligase, to initiate mammalian pexophagy. However, a recent study suggested that increased cytoplasmic ROS can stimulate ubiquitination, allowing PEX5 to act as a ROS sensor, subsequently causing pexophagy [168,169,170]. After PEX5 is ubiquitinated, it is recognized by NBR1 or p62, which binds to the autophagosome, and pexophagy occurs [43,167,171].

In certain neurodegenerative diseases, the number and function of peroxisomes may be impaired. In AD, Aβ and tau proteins aggregate in the neurons, and the peroxisomes may be affected. One study has shown that treatment with a peroxisome proliferator, Wy-14.463, in an Aβ overexpressing rat hippocampus, increased peroxidase numbers and catalase activity, reduced ROS production, and produced the denaturing effects of beta-amyloid [172]. In a clinical study, plasma glycogen decreased in postmortem brain samples from AD patients, indicating reduced peroxisome activity or a shortened half-life of plasmalogens [173]. Abnormally reactive peroxisomes lead to excess ROS production, which, in turn, leads to activation of ATM kinase, which promotes phosphorylation and subsequent autoubiquitination of PEX5 to initiate pexophagy [169]. Therefore, as an important mechanism of neurodegenerative diseases, pexophagy may be beneficial in neurodegenerative diseases. However, there is no direct evidence that pexophagy and AD are related. Another gap in the literature is that there are no sex- or age-related studies on peroxisomes in neurological disorders. For example, there are studies on sex differences in response to cerebral ischemia or ischemic stroke, but it is unclear how these sex-related differences specifically affect peroxisomes [174,175,176]. Considering the effects of the peroxisome in the brain, future research on age-related neurological changes should investigate how the peroxisome pathways are affected, particularly in AD.

## 7. Nucleophagy

Nucleophagy refers to the selective removal of nuclear components by autophagy [177,178]. Mutations in Lamin A/C and Emerin lead to the formation of giant autophagosomes for the clearance of damaged nuclear material, thereby causing envelopathy [179]. The degradation of LMNB1 (Lamin B1) and chromatin during oncogene-induced human cell senescence is mediated by autophagy directly interacting with MAP1LC3B, suggesting the existence of a protective mechanism against tumors [180]. Although there is no evidence of a link between nucleophagy and AD, studies have found that LMNB1 mutation causes human degenerative disease and adult demyelinating leukodystrophy, and that LMNB1 and MAP1LC3B must interact for nucleophagy to occur [180,181,182]. Activation of LMNB1 can significantly delay cell senescence and alleviate degenerative diseases, suggesting that LMNB1-induced nucleophagy may have a protective effect [183,184,185].

## 8. Lysophagy

Lysophagy refers to damaged lysosomes being encapsulated by autophagosomes and fused with healthy lysosomes through autophagy. At the same time, β-galactoside is released from the damaged lysosome and is sensed by LGALS3/GAL3 (galectin 3), a key marker of lysosomal phagocytosis [186]. In addition, TRIM16, the E3 ligase of the TRIM family, is recruited to LGALS3 and localized to the damaged lysosomes, where it is further ubiquitinated to initiate autophagy, thereby removing the damaged lysosomes [187,188]. Similar to other types of organelle-selective autophagy, it relies on the degradation of p62-MAP1LC3B ubiquitination [186,189]. A novel ubiquitinated glycoprotein, FBXO27, also regulates impaired lysosomal recruitment [190]. In AD models, when lysophagy is damaged, the accumulated proteins cannot be cleared in a timely manner, which aggravates the symptoms of AD. It has been found that Aβ promotes the activation of NLRP3 inflammasomes by inducing lysosomal leakage; this indicates that lysosomal abnormalities in AD lead to inflammatory responses [191]. Moreover, lysosomal dysfunction can lead to neurodegeneration, and some lysosome-related gene mutations, such as valosin-containing protein (VCP), p62, OPTN, or TBK1, can cause lysosomal damage. These mutations have also been confirmed to be involved in the pathogenesis of neurodegenerative diseases [192,193,194]. Therefore, quality control of the lysosomes is crucial in neurodegenerative diseases.

## 9. Ribophagy

Ribosomes are the main organelles responsible for protein synthesis and translation; hence, they are crucial to cell homeostasis [195]. Generally, ribosome turnover in the cell is controlled by a selective autophagy process, known as ribophagy. Studies have shown that, in yeast, the ubiquitin protease ubiquitin carboxyl-terminal hydrolase 3 (UBP3) and its cofactor, UBP3-associated protein BRE5 (Bre5), participate in this process, which mainly targets the 60S subunit but does not affect the 40S subunit [196]. The recent discovery that USP10 and G3BP1 are homologous genes of UBP3 and Bre5 in mammals suggests that ribophagy is evolutionarily conserved [197]. There is increasing evidence that in mammalian cells, ribosomes are consumed by autophagosomes. Ribosomes are also involved in neurodegeneration, indicating that ribophagy plays a key role in cell function and vitality [198]. Studies have shown that during nutrient starvation, NUFIP1, an LIR motif-containing ribosome receptor, binds to LC3B on autophagosomes, mediates ribosome turnover, and promotes cell survival.

## 10. Therapeutic Strategies for AD Based on Selective Autophagy

Although the detailed mechanism(s) of selective autophagy in AD remains to be elucidated, enhancing autophagy by genetic manipulation or pharmacological approaches has been beneficial in different AD models. Overexpression of PINK1 restores mitochondrial function, reduces Aβ production and amyloid pathology, and attenuates synaptic and cognitive function by activating mitophagy signaling in APP transgenic mice [199]. In addition, AD is an aging-related disease and aging damages organelles, especially the mitochondria, causing the accumulation of abnormal proteins that damage the quality control of organelles. This undoubtedly plays a great role in the survival and function of neurons, and the clearance of these damaged organelles and proteins will benefit neuron homeostasis. PPAR/PPARα is a transcription factor that regulates neural autophagy in the system, and its mediated autophagy can affect AD. In one study, its agonist, Wy14643, was found to activate autophagy, reduce Aβ protein levels, and treat cognitive impairment in AD mice [200]. In this process, PPAR/PPARα is a peroxisome-related receptor, but it can regulate lipid metabolism-related genes, thereby regulating autophagy in the nervous system, which indicates that in cells, there is a crosslink between different organelles. In AD mice and nematode models, mitochondrial stimuli, such as NAD^+^ supplementation, urolithin A, and actinonin, increased PINK1 and Parkin-dependent mitophagy and alleviated AD pathology, suggesting that the removal of damaged mitochondria can act as a potential therapeutic strategy [201]. Furthermore, studies have found that microRNA can treat AD by enhancing autophagy and improving synaptic function [202]. However, it is not clear whether these microRNA participate in selective autophagy. Moreover, in the latest research, it was found that damage to the lysosome can cause synaptic dysfunction, which can cause nervous system-related diseases [203]. Lysosomes are the most critical degradation centers for all kinds of autophagy, so clearing damaged lysosomes and maintain healthy lysophagy quality will play an important role in AD treatment. A study found that impaired CMA caused neuronal dysfunction, increased protein toxicity, and neurological disorders, while chemical enhancement of CMA delayed the onset of AD in mice model [204]. This also implies that we can analyze its structure and function according to the CMA-specific receptors, find specific targets, conduct drug development, and specifically enhance the function of the CMA, which is a strategy for treating AD.

## 11. Discussion

Current studies have indicated that multiple autophagies may together contribute to the pathogenesis of AD. An increase in aggregating proteins will cause cytotoxicity and damage mitophagy, while damage to the mitochondria will aggravate the pathology of AD, forming a vicious cycle and causing damage to the body. Impaired mitochondrial function can also lead to changes in other organelles, such as reduced energy synthesis, reduced endoplasmic reticulum protein processing, and decreased synthesis of degrading enzymes, resulting in the accumulation of damaged proteins [205]. Studies have found that reticulophagy may play a protective role in AD patients, and the accumulation of toxic proteins can lead to positive feedback regulation of the ER, which degrades the damaged ER and misfolded proteins through macroautophagy and CMA.

In the subtypes of selective autophagy, the identification of AD-related specific receptors is a big challenge. For aggrephagy, several receptors have been confirmed, such as p62, NBR1, ALFY, and OPTN [206]. However, we need to further determine whether the degradation of Aβ and PHF-tau aggregates, and TNF are receptor-specific. If so, drugs that target specific receptors can be discovered and studied to promote degradation, which would then become a viable strategy for treating AD. For mitophagy, post-translational modifications (PTMs) are regarded as a key mechanism [207]. In addition, PINK1, BNIP3L, BNIP3, FUNDC1, NIPSNAP1, NIPSNAP2, BCL2L13, prohibitin 2, MCL-1, OPTN, and FKBP prolyl isomerase 8 (FKBP8) [208,209] have been identified as receptors. However, only a few receptors have been reported to be involved in the pathogenesis of AD. Are there other mitophagy receptors that mediate the clearance of damaged mitochondria in AD? With regard to reticulophagy and lipophagy, while there is very little evidence of their involvement in the pathogenesis of AD, ER stress, ER injury, and mutations in the reticulophagy-related receptor RTN3 have been demonstrated to be associated with AD pathogenesis [210]. Enhanced lipophagy can reduce the accumulation of lipid droplets and thereby reduce the neuronal neurodegeneration caused by an accumulation of dihydroceramide desaturase [211]. Therefore, more evidence is needed to support the involvement of reticulophagy and lipophagy in the pathogenesis of AD. Overall, the discovery of specific clearance mechanisms for these damaged organelles will provide new targets for the treatment of AD. Here, we have listed the related receptors in selective autophagy in AD (Table 1).

In macroautophagy, substrates are recognized by different receptors in different selective autophagy pathways for degradation. Furthermore, at least 30% of the protein degradation process is through CMA, which is also lysosome-dependent and highly selective. The misfolded protein is further identified by Hsc70 by attaching a KFERQ motif, and is degraded by the lysosomal receptor Lamp2A. How does the cell recognize and maintain its specificity? This needs to be studied further, but it gives us a new way to focus on the receptor. Is it the structure of the receptor that determines this process and is there a propensity to degrade the damaged protein under certain conditions? In one recent study, APP protein was found to be the receptor of CMA [240], and APP could conduct both macroautophagy and CMA. However, how to conduct macroautophagy and CMA, and whether there is a priority between the two are problems to be solved in the future.

Finding the specific receptors will be the key point for promoting selective autophagy in AD. Therefore, further studies to find other autophagy-related receptors, develop specific drugs, or discover crosslinks between different subtypes of selective autophagy should provide new approaches for the treatment of AD.

## Figures and Tables

**Figure 1 ijms-23-03609-f001:**
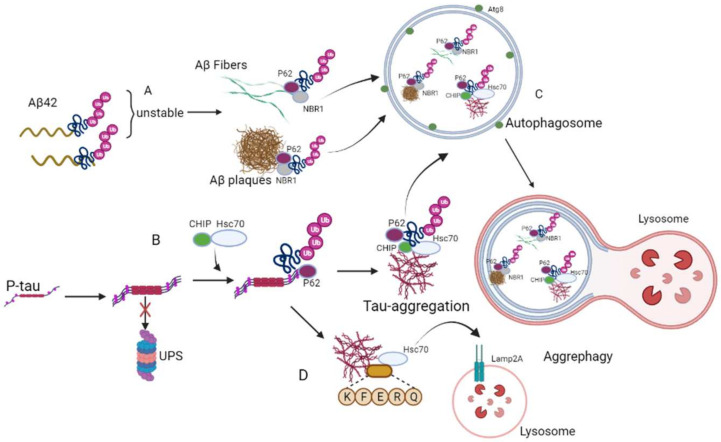
Mechanism of aggrephagy during the occurrence of AD. (**A**) Polyubiquitinated Aβ plaques and fibers binds to p62 and NBR1 specifically, and initiate degradation through aggrephagy. (**B**) Hsc70 interacts with CHIP to regulate tau ubiquitination and then is recruited by p62. (**C**) All these aggregate proteins are wrapped by autophagosomes for degradation. (**D**) KFERQ binds with tau aggregates and is transferred to the lysosome by the Lamp2A receptor for degradation.

**Figure 2 ijms-23-03609-f002:**
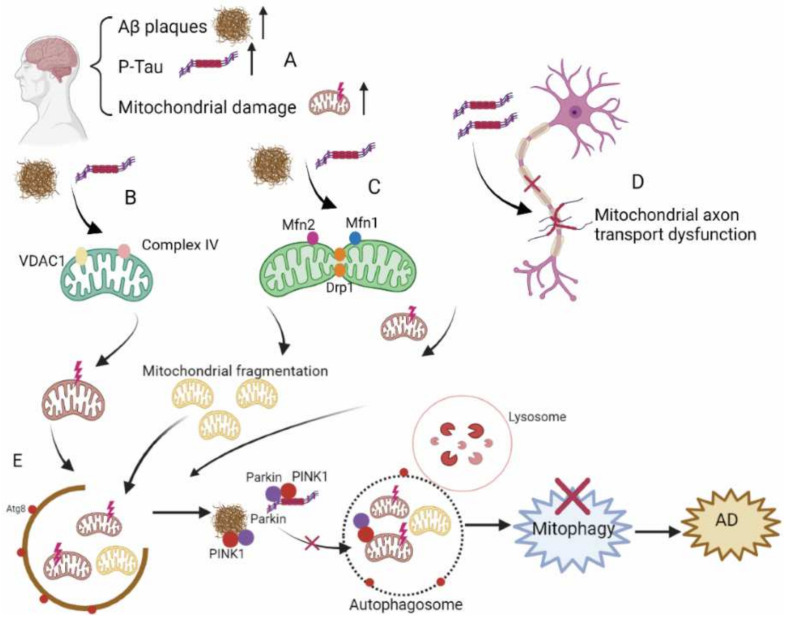
Mitophagy in the pathogenesis of AD. (**A**) Impairment of the mitochondria occurs in the early stages of AD, and Aβ and P-tau are increased. (**B**) Aβ and P-tau interact with VDAC1 and Complex IV, and interfere with mitochondrial function. (**C**) Aβ and P-tau induce mitochondrial fragmentation by reducing MFN1/2 and activating Drp1. (**D**) Hyperphosphorylation of tau blocks the binding of tau with microtubules, disrupting mitochondrial axon transport. (**E**) Aβ and P-tau disturb the recruitment of PINK1 and Parkin, causing mitophagy dysfunction.

**Figure 3 ijms-23-03609-f003:**
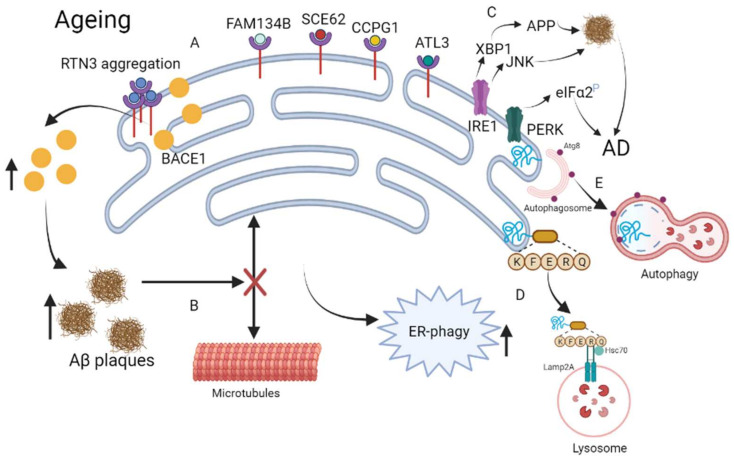
Reticulophagy in the pathogenesis of AD. (**A**) RTN3 aggregation counteracts the negative regulation of BACE1 activity by RTN3, causing Aβ to increase in the aging brain. (**B**) Deposition of Aβ inhibits the interaction of the ER and microtubules, inducing reticulophagy. (**C**) ER stress/AD factors increase misfolded proteins through the IRE1 and PERK pathway. (**D**) KFERQ binds with impaired ER for CMA degradation. (**E**) Autophagosomes wrap the impaired ER for degradation.

**Table 1 ijms-23-03609-t001:** List of selective autophagy receptors in the pathogenesis of AD.

Selective Autophagy	Receptor	References	AD-Related Receptor
Aggrephagy	P62; NBR1; ALFY Hsc70; CHIP OPTN; TOLLIP TAX1 binding protein 1	[38,39] [57] [212,213] [214]	P62; NBR1; ALFY Hsc70; CHIP
Mitophagy	PINK1/Parkin VDAC; RHOT1 MFN1/2; BNIP3L FUNDC1; BNIP3 AMBRA1; BCL2LI3 FKBP8; CHDHD ISC1; PHB2 Cardiolipin; NIPSNAP1/2 NDP52; OPTN MCL-1; P62	[201,215] [216,217] [218,219] [208,220] [221,222] [223,224] [225,226] [227,228] [229,230] [231,232]	PINK1/Parkin MCL-1; DISC1
Reticulophagy	FAM134B.; SEC62 CCPG1; RTN3 ATL3	[233,234] [32,210,235] [236]	RTN3
Lipophagy	ATGL	[144]	-
Pexophagy	NBR1; P62; ABCD3	[167] [237]	-
Nucleophagy	Atg39	[238]	-
Lysophagy	Galectin 3	[186]	-
Ribophagy	NUFIP1	[239]	-

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
