# Peer review of "Mechanistic Insights into Selective Autophagy Subtypes in Alzheimer’s Disease"

_ijms, 2022, doi:10.3390/ijms23073609_

Round 1

Reviewer 1 Report

Content is well curated and comprehensive, some sentences require review and rephrasing.  Pay extra attention to Picture 1.  All comments can be found in the attached PDF.

Author Response

Point-by-point rebuttal letter to the expert reviewers

We thank the reviewers for providing the valuable comments for revision. We have addressed each comment carefully and made revisions accordingly. The point-by-point response to the reviewers’ comments are listed as following:

Reviewer-1

Response: We have fixed the typos as per the comments of Reviewer 1.

Reviewer 2 Report

The manuscript performed by Guan et al. aims to describe the different types of selective autophagy mechanisms involved in AD progression.

The manuscript is well written and clearly organized, however, several important issues should be addressed:

Major concern

This manuscript is purely descriptive and it does not contribute to the current knowledge. The only interesting paragraph is the last one of the discussion section. In this regard, as a suggestion, the current manuscript should be shortened to describe all the different specific and selective autophagy subtypes, then, a paragraph regarding the different crosslinks between the different autophagy types (also including macro, micro and CMA) in AD should be addressed. Lastly, another paragraph must be added to address the recent studies that have tried to modulate these pathways and to discuss the obtained results. That would make a profit for the development of new strategies for AD therapy.

Other small issues

Figures:

Figure 1 needs to be completed; it misses the KFERQ motif, LAMP2A receptor in the autophagosome membrane, Atg8…

Same applies for figure 2 and 3, especially in figure 3 where ER-associated autophagosomes and lysosomes, and UPR are not even drawn.

Typos:

  • Line 21: please, add “,” between nucleophagy and lysophagy.
  • Line 23: pathogenesis without capital letter.
  • Line 94: please correct as follows; “is IN correlation WITH AD pathogenesis”.
  • Line 95: please correct as follows; “Aggrephagy has”.
  • Sentence lines 160 and 161 is missing a verb.
  • Line 185: please correct as follows; “many research HAS shown”
  • Line 252: comas
  • Line 256: conditions

Author Response

Point-by-point rebuttal letter to the expert reviewers

We thank the reviewers for providing the valuable comments for revision. We have addressed each comment carefully and made revisions accordingly. The point-by-point response to the reviewers’ comments are listed as following:

Reviewer-2

Major concern

This manuscript is purely descriptive and it does not contribute to the current knowledge. The only interesting paragraph is the last one of the discussion section. In this regard, as a suggestion, the current manuscript should be shortened to describe all the different specific and selective autophagy subtypes, then, a paragraph regarding the different crosslinks between the different autophagy types (also including macro, micro and CMA) in AD should be addressed. Lastly, another paragraph must be added to address the recent studies that have tried to modulate these pathways and to discuss the obtained results. That would make a profit for the development of new strategies for AD therapy.

Response: Paragraphs describing selective autophagy subtypes are shortened. The crosslinks between the different autophagy types were discussed in revised discussion section. We have added a review of AD therapeutic strategies and benefits based on selective autophagy.

Other small issues

Figures:

Figure 1 needs to be completed; it misses the KFERQ motif, LAMP2A receptor in the autophagosome membrane, Atg8…

Response: In revised Figure 1, we have added KFERQ motif, LAMP2A receptor on the lysosomal membrane and Atg8 on the autophagosome double membrane in accordance with the reviewer’s suggestion.

Same applies for figure 2 and 3, especially in figure 3 where ER-associated autophagosomes and lysosomes, and UPR are not even drawn.

Response: In the revised Figure 2 and 3. We have included the ER-associated autophagosomes and lysosomes. 

Typos:

Line 21: please, add “,” between nucleophagy and lysophagy.

Line 23: pathogenesis without capital letter.

Line 94: please correct as follows; “is IN correlation WITH AD pathogenesis”.

Line 95: please correct as follows; “Aggrephagy has”.

Sentence lines 160 and 161 is missing a verb.

Line 185: please correct as follows; “many research HAS shown”

Line 252: comas

Line 256: conditions

Response: We have fixed the typos as per the comments of Reviewer 2.

Round 2

Reviewer 2 Report

The manuscript has been properly improved, and now it offers new questions and problems that need to be addressed to further develop new strategies for AD therapy.